# FREE-GRAINED HIERARCHICAL RECOGNITION

## ABSTRACT

Hierarchical image classification predicts labels across a semantic taxonomy, but existing methods typically assume complete, fine-grained annotations, an assumption rarely met in practice. Real-world supervision varies in granularity, influenced by image quality, annotator expertise, and task demands; a distant bird may be labeled *Bird*, while a close-up reveals *Bald eagle*. We introduce ImageNet-Free, a large-scale benchmark curated from ImageNet and structured into cognitively inspired basic, subordinate, and fine-grained levels. Using CLIP as a proxy for semantic ambiguity, we simulate realistic, mixed-granularity labels reflecting human annotation behavior. We propose *free-grain learning*, with heterogeneous supervision across instances. We develop methods that enhance semantic guidance via pseudo-attributes from vision-language models and visual guidance via semi-supervised learning. These, along with strong baselines, substantially improve performance under mixed supervision. Together, our benchmark and methods advance hierarchical classification under real-world constraints.

## 1 INTRODUCTION

Hierarchical classification (Chang et al., 2021; Chen et al., 2022; Jiang et al., 2024; Park et al., 2025) predicts a **semantic tree** of labels, capturing categories from broad to specific. This richer output supports flexible use: An expert may seek *Bald Eagle*, while a general user may only need *Bird*. Moreover, predicting the full hierarchy improves robustness and scalability, encouraging models to generalize across levels, and can naturally support extensions like adding new parent or child classes.

However, existing methods (Chang et al., 2021; Wang et al., 2023) assume *complete supervision* at all levels for all the training examples, which rarely holds in practice. In real-world settings, annotation granularity depends on image clarity, annotator expertise, or task-specific needs: A distant bird could only be labeled as *Bird*, while a close-up allows *Bald eagle* (Fig.1).

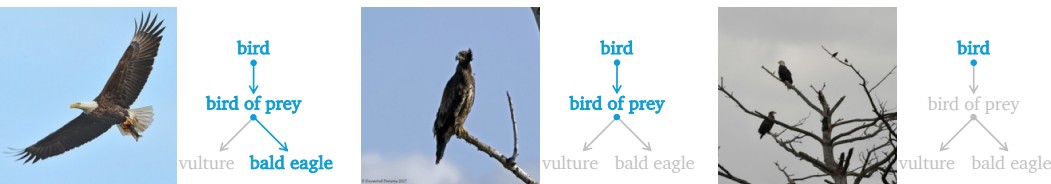

Figure 1: **Images vary in semantic detail: Some support only coarse labels, others reveal fine-grained categories.** We propose *free-grain learning*: Training a hierarchical classifier with *supervision free to vary in granularity across examples*, reflecting semantic ambiguity in real-world images.

We propose **free-grain learning**, where supervision is free to vary in granularity: Training labels may appear at any level of a fixed taxonomy, e.g., *Bird*, *Bird of prey*, or *Bald eagle*. The key challenge is to predict the full taxonomy from training data with **mixed** (rather than *uniform fine-grained*) labels. This task not only reflects real-world variability in annotation quality and specificity, but also enables learning from partially labeled data at scale. It further requires integration **across semantic annotation granularities and across visual instances**, as the model must infer a *complete taxonomy* for each example based on **heterogeneous supervision**.

However, existing benchmarks are ill-suited for this task (Table 1). Small datasets such as CUB (Welinder et al., 2010) and Aircraft (Maji et al., 2013) lack scale, while iNaturalist (Van Horn

Table 1: **Existing Hierarchical Recognition Benchmarks Are Insufficient.** CUB (Welinder et al., 2010) and Aircraft (Maji et al., 2013) provide clean hierarchies but are small; iNat21-mini (Van Horn et al., 2021) has a clean taxonomy but is restricted to biology; ImageNet (Russakovsky et al., 2015) is large but structurally inconsistent. We introduce *ImageNet-Free*, a general-purpose large-scale benchmark with a coherent three-level hierarchy grounded in cognitive psychology. We also provide Free-grain variants of CUB, Aircraft, and iNat21-mini for broader evaluation.

| Dataset | #levels | #classes | #train | #test |
|---|---|---|---|---|
| CUB | 3 | 13-38-200 | 5,994 | 5,794 |
| Aircraft | 3 | 30-70-100 | 6,667 | 3,333 |
| iNat21-mini | 8 | 3-11-13-51-273-1103-4884-10000 | 500,000 | 100,000 |
| ImageNet | 5-19 | - 1000 | 1,281,167 | 50,000 |
| **ImageNet-Free** | **3** | **20-127-505** | **645,480** | **25,250** |

et al., 2021) is limited to biology and unsuitable for general-purpose evaluation. Larger benchmarks like ImageNet (Russakovsky et al., 2015) and tieredImageNet (Ren et al., 2018) inherit noisy, inconsistent hierarchies from WordNet (Fellbaum, 1998). As shown in Fig. 2, hierarchy depths vary widely from 5 to 19 levels, with some classes following multiple paths—for example, *Minivan* appears in four different paths (depths 12–15), while *Teddy bear* appears only once at depth 7. Such inconsistencies make evaluation ambiguous: one fine class can map to several hierarchies, and predictions often traverse long chains of redundant nodes (e.g., *entity*, *object*). As a result, most methods on ImageNet and tieredImageNet restrict evaluation to **leaf-node accuracy**, with auxiliary metrics like mistake severity (Bertinetto et al., 2020; Garg et al., 2022b; Jain et al., 2023).

To address these limitations, we construct ImageNet-Free, a benchmark with a well-structured three-level hierarchy: basic (e.g., *Dog*), subordinate (e.g., *Shepherd*), and fine-grained (e.g., *German Shepherd*) (Fig. 3). Grounded in cognitive psychology (Rosch et al., 1976; Rosch, 1978) and folk taxonomies (Berlin et al., 1966), our design reflects that the basic level is the most natural and widely recognized category for humans, while subordinate and fine-grained levels capture increasingly specific distinctions. By focusing on this range—from the most intuitive to the most detailed—we enable semantically meaningful hierarchical prediction, avoiding abstract or redundant levels (e.g., *Physical Entity* in the original ImageNet hierarchy) that provide little practical value.

To simulate mixed-granularity labeling, we use CLIP (Radford et al., 2021) as a proxy for visual-semantic ambiguity or annotator errors, and determine which levels of labels to retain based on its predictions. We observe that this produces realistic distributions: *e.g.*, distant shots labeled as *Bird*, mid-range views as *Bird of prey*, and close-ups as *Bald eagle* (Fig. 4). Our final dataset consists of 645,480 images across 20 basic, 127 subordinate, and 505 fine-grained classes (Table 1). We apply the same approach to iNat21-mini and CUB using BioCLIP (Stevens et al., 2024), a foundation model for biology, producing iNat21-mini-Free and CUB-Free. We also construct synthetic CUB-Rand and Aircraft-Rand to enable controlled evaluation under varying label sparsity and granularity.

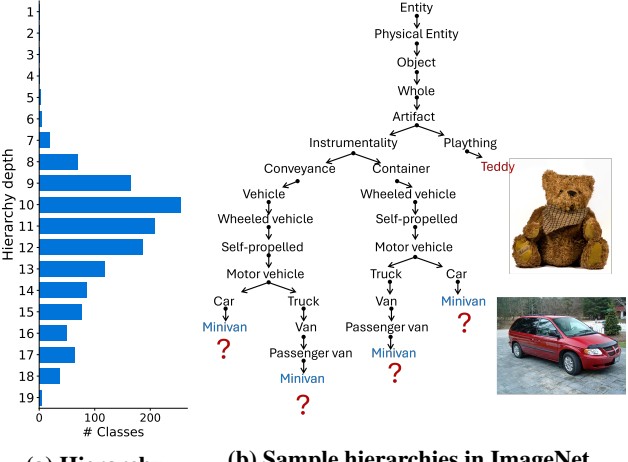

**(a) Hierarchy depths**

**(b) Sample hierarchies in ImageNet** (*Minivan & Teddy*)

Figure 2: **Inconsistent and Noisy Hierarchy of ImageNet WordNet.** (**a**) The histogram of hierarchy depths shows that ImageNet classes range from 5 to 19 levels, with many exceeding 10, which hinders consistent evaluation. (**b**) Sample hierarchies illustrate that classes can have multiple paths of different depths: *Minivan* appears in four paths at depths 12–15, while *Teddy bear* exists only at depth 7. This imbalance and inconsistency in the hierarchy make it unclear which path should be considered correct, underscoring the difficulty of using the original WordNet hierarchy for training and evaluation.

**(a)** Original ImageNet's WordNet hierarchy **(b)** Our 3-level hierarchy

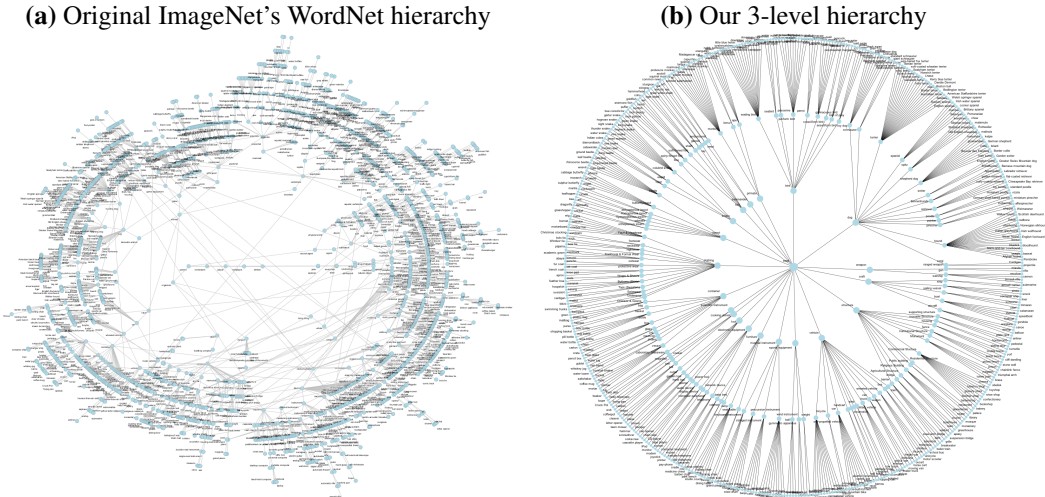

Figure 3: **We curate ImageNet-Free as a benchmark for free-grain hierarchical classification.** **(a)** The original ImageNet taxonomy is noisy and inconsistent, with imbalances, overlaps, and multiple paths, making it unsuitable for reliable evaluation. **(b)** We construct a coherent 3-level taxonomy, inspired by cognitive psychology (Rosch et al., 1976): *basic* for general recognition, *subordinate* for contextual specificity, and *fine-grained* for specialized distinctions.

When applied directly under free-grain setting, existing hierarchical classifiers (Chen et al., 2022; Park et al., 2025) degrade severely—up to **–40%** full-path accuracy on iNat21-mini —highlighting the difficulty of the task. To address this, we propose three additional strategies: **1)** learning pseudo-

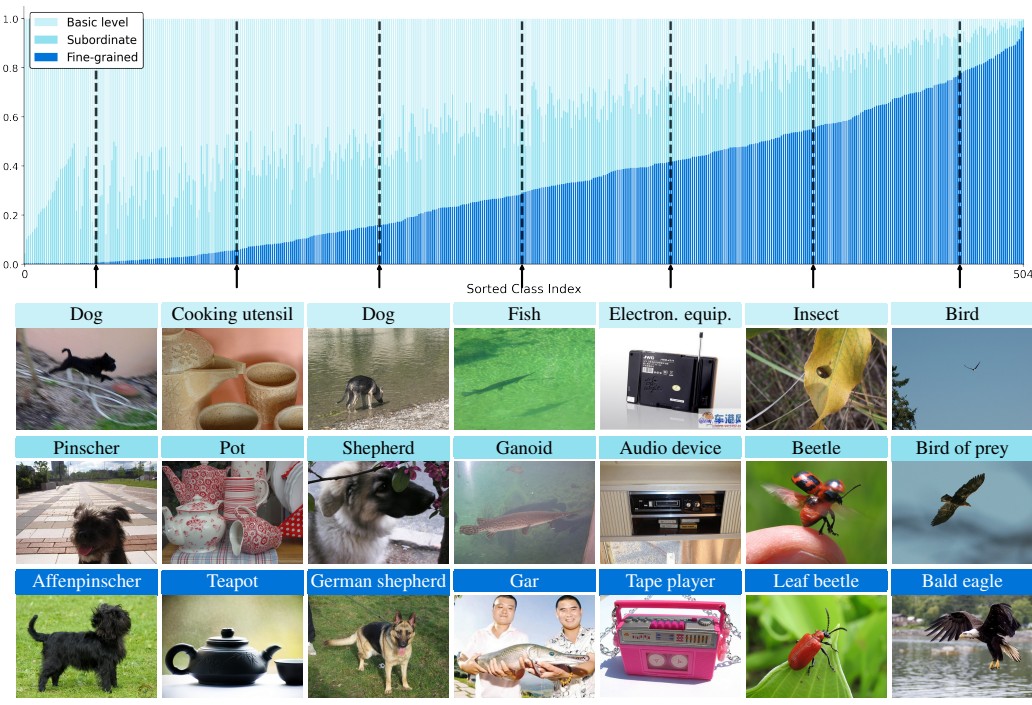

Figure 4: **Our ImageNet-Free captures real-world challenges, where fine-grained labels follow a long-tailed distribution and granularity varies with visual clarity. Top:** The graph shows the proportion of basic, subordinate, and fine-grained labels per class (sorted by ID). Fine-grained labels are scarce on the left but increase toward the right, forming a long tail. This imbalance often causes models to overfit to basic-level features and miss subtle distinctions, underscoring the need for robust multi-level learning. **Bottom:** Arrow-marked samples illustrate how label pruning reflects difficulty. (Last Column): a distant bird is labeled at the basic level (Bird); one with visible wings and talons at the subordinate level (Bird of prey); and a close-up at the fine-grained level (Bald eagle).

attributes (e.g., short legs, docked tail) from vision–language models to provide semantic cues when finer labels are missing; **2)** applying semi-supervised learning by treating missing-grain labels as unlabeled; **3)** combining both approaches. Across datasets, these methods outperform hierarchical baselines by +4–25%p, establishing stronger baselines for free-grain learning.

**Contributions. 1)** We introduce free-grain learning for hierarchical classification, capturing real-world variability in label granularity. **2)** We present ImageNet-Free, with a cognitively grounded 3-level hierarchy, and additional free-grain benchmarks across diverse domains. **3)** We establish strong baselines that significantly improve performance by leveraging semantic and visual guidance.

## 2 RELATED WORK

**Hierarchical classification** has been studied mainly for leaf-node prediction on large but inconsistent taxonomies such as ImageNet (Karthik et al., 2021; Zhang et al., 2022), or for full-taxonomy prediction on small datasets like CUB and Aircraft (Chang et al., 2021; Park et al., 2025). These settings lack the scale, diversity, and label sparsity needed for realistic evaluation. Our work instead enables full taxonomy prediction under heterogeneous supervision on large-scale data.

**Imbalanced and semi-/weakly-supervised classification** have been widely explored (Liu et al., 2019; Tarvainen & Valpola, 2017; Robinson et al., 2020), but mostly at a single fine-grained level or with fully observed coarse labels. In contrast, we address both intra- and inter-level imbalance, requiring consistent prediction across multiple granularities with partially missing supervision. See a full task comparison in Table 2.

**Foundation models** such as CLIP (Radford et al., 2021) has been used for zero-shot flat classification via text prompts (Pratt et al., 2023; Saha et al., 2024). In contrast, our approach leverages text only during training to learn visual patterns across levels, requiring no textual input at inference.

Further discussion and additional related work are provided in Appendix B.

Table 2: **Our task setting is more practical and challenging than existing ones.** Ours reflects realistic scenarios where annotations are free-grain and imbalanced, requiring hierarchical predictions to balance accuracy and consistency across levels.

| Tasks | Input Labels | Output Labels | Training Labels | Label imbalance | Evaluation Metrics |
|---|---|---|---|---|---|
| Hierarchical Classification | Hierarchical | Hierarchical | Fully annotated, all levels | $\times$ | Accuracy & Consistency |
| Imbalanced Classification | Fine-grained | Fine-grained | Fully annotated, single level | Intra-level | Accuracy |
| Semi-supervised Learning | Fine-grained | Fine-grained | Missing labels, single level | $\times$ | Accuracy |
| Weakly-supervised Classification | Coarse-grained | Fine-grained | Fully annotated, single level | $\times$ | Accuracy |
| **Free-Grain Learning** | Hierarchical | Hierarchical | Missing labels, all levels | Intra- & Inter-level | Accuracy & Consistency |

## 3 HIERARCHICAL DATASET FOR FREE-GRAINED RECOGNITION

**3.1 Defining Three-Level Taxonomy for ImageNet-Free.** We restructure ImageNet (Russakovsky et al., 2015)'s WordNet (Fellbaum, 1998)-based hierarchy into a consistent three-level taxonomy, explicitly guided by Rosch's categorization principles (Rosch et al., 1976). In Rosch's framework, the *basic level* (e.g., *dog*, *car*) is the most natural and visually distinctive, balancing generality and specificity; it is also the level people most often use in everyday recognition and naming, unlike abstract superordinate categories (e.g., *animal*) or overly narrow subordinate ones (e.g., *Pembroke*).

We adopt the basic level as the coarsest node in each branch, with *subordinate* and *fine-grained* levels (e.g., *Corgi → Pembroke*) providing progressively finer distinctions. However, WordNet chains such as *artifact → ... → vehicle → ... → motor vehicle → car → ambulance* can yield only two usable levels if *car* is taken as basic. In these cases, we elevate Rosch's superordinate category (e.g., *vehicle*) to serve as the basic level, which remains visually distinctive from other basic categories (e.g., *craft*, *container*) and ensures a three-level hierarchy. This yields branches that support three semantically coherent and visually meaningful levels for hierarchical prediction.

Specifically, we adopt the following systematic principles: **1)** *Enforce meaningful structure:* We remove paths where each node has only one child, since coarse labels fully determine the fine labels. Branches with fewer than three levels are also excluded. **2)** *Maximize within-group diversity:* Among subordinate candidates under each basic class, we favor those with richer fine-grained subclasses—e.g., *parrot* (4 children) over *cockatoo* (1 child). **3)** *Refine vague categories:* Ambiguous

groups such as *Women's Clothing* are reorganized into precise, functionally grounded categories (e.g., *Underwear*) to improve clarity. **4) *Validate with language models and human review:*** We use language models (ChatGPT (Achiam et al., 2023)) to suggest refinements, with all decisions manually reviewed for semantic consistency. Applying this curation process to ImageNet-1k yields a structured benchmark of 20 basic, 127 subordinate, and 505 fine-grained classes, ensuring every branch supports meaningful hierarchical prediction (a complete list is provided in Appendix A).

### 3.2 Semantic Label Pruning for ImageNet-Free, iNat21-mini-Free, and CUB-Free

To build a realistic free-grain training dataset, we prune hierarchical labels using large vision–language models as a proxy for visual–semantic ambiguity: CLIP (Radford et al., 2021) for ImageNet-Free and BioCLIP (Stevens et al., 2024) for iNat21-mini-Free and CUB-Free. Although these models are not explicitly designed to measure ambiguity, their zero-shot confidence consistently correlates with visual distinctiveness (Fig. 4). Moreover, since label annotation is affected by annotator expertise or error, this proxy offers a practical approximation.

We adopt CLIP's prompt-ensemble strategy (e.g., *a photo of a [class]*, *art of a [class]*) and compute average confidence for fine-grained and subordinate levels. Labels are retained based on prediction correctness: **(1)** If both fine-grained and subordinate are correct, we keep all labels. **(2)** If only subordinate is correct, we keep up to that level. **(3)** Otherwise, only the basic label is kept. We further prune subordinate labels proportionally to the fine-grained removal rate per class.

**(1) ImageNet-Free.** After pruning, 32.6% of images retain all three levels (Basic + Subordinate + Fine-grained), 28.0% retain two (Basic + Subordinate), and 39.4% retain only the Basic. Each class maintains the same number of images as ImageNet; imbalance arises only from label granularity. **(2) iNat21-mini-Free.** Although BioCLIP is trained on iNat21-mini 's full taxonomy, it performs well when predicting fine-grained species but struggles when restricted to coarser labels. This gap enables substantial pruning: 22.5% of images retain all three levels (Order + Family + Species), 28.0% retain two, and 49.5% retain only Order. **(3) CUB-Free.** With the same procedure, 31.5% of images keep three levels, 23.3% two (Order + Family), and 45.2% only Order.

### 3.3 Synthetic Label Pruning for CUB-Rand and Aircraft-Rand

To control label availability, we construct synthetic variants—CUB-Rand and Aircraft-Rand —by randomly pruning labels from CUB (Welinder et al., 2010) and Aircraft (Maji et al., 2013). Unlike realistic pruning, this design systematically varies supervision and simulates *extreme* sparsity (e.g., only 10% fine-grained labels), enabling stress-testing of model robustness across diverse label distributions. Although random removal is independent of image difficulty, it reflects practical factors such as annotator expertise, cost, or task-specific constraints. We denote availability as *a-b-c*, where $a\%$ of basic, $b\%$ of subordinate, and $c\%$ of fine-grained labels are retained (e.g., 100-50-10 retains 10% fine-grained labels and 40% subordinate-only labels).

## 4 Free-Grain Learning Methods for Hierarchical Classification

**4.1 Problem setup.** We begin by describing the problem setup. In free-grain hierarchical classification, the goal is to train a model that predicts object categories across all levels of a taxonomy, given training data with labels of varying granularity. Formally, let $\mathcal{X}$ denote the input space of images, and $\mathcal{Y}_1, \ldots, \mathcal{Y}_L$ the label spaces at $L$ hierarchical levels, from coarsest ($\mathcal{Y}_1$) to finest ($\mathcal{Y}_L$). Each training sample consists of an image $x \in \mathcal{X}$ and a partial label set $\{y_l\}_{l \in \mathcal{S}_x}$, where $\mathcal{S}_x \subseteq \{1, \ldots, L\}$ indicates the levels at which labels are provided.

We assume that if a label $y_l$ is available, all coarser labels $y_{l'}$ for $l' < l$ are also available due to the structure of the taxonomy, while finer labels $y_{l'}$ for $l' > l$ are missing. We further assume the coarsest label $y_1$ is always given. The objective is to learn a classifier $f : \mathcal{X} \rightarrow \mathcal{Y}_1 \times \cdots \times \mathcal{Y}_L$, $f(x) = (\hat{y}_1, \ldots, \hat{y}_L)$, that predicts labels at all levels of the hierarchy.

**4.2 Baselines.** With no existing baselines for this new setting, we propose four strong baselines, each approaching the problem from a different perspective.

**(1) Semantic Guidance: Text-Guided Pseudo Attributes (Text-Attr).** Our semantic guidance approach is motivated by the observation that while class labels differ across hierarchical levels (e.g., *Dog → Corgi → Pembroke*), many visual attributes—such as tail length or ear shape—remain consistent (Fig. 5). To capture these shared semantic cues, we use image descriptions as auxil-

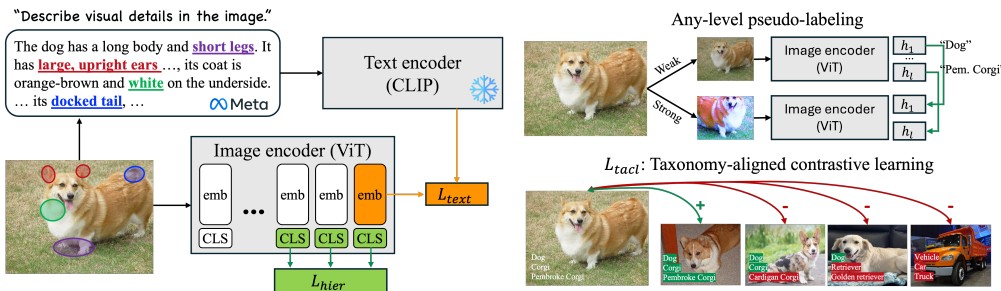

Figure 5: **Text-Attr enriches feature representations using semantic cues from images, compensating for missing labels and capturing shared attributes across levels.**

Figure 6: **Taxon-SSL handles missing-level labels by treating them as unlabeled and learns from visual consistency through augmented views.**

iary supervision. Instead of class-name prompts for zero-shot classification, we extract free-form descriptions directly from the image, independent of labels.

Specifically, given an input image $x$, we use a frozen vision-language model (VLM), Llama-3.2-11B (Dubey et al., 2024), to generate a language description $d_x$, using the prompt: "*Describe visual details in the image.*" This produces descriptions containing phrases such as "short legs" or "pointed ears," which we encode into a text embedding $z_x^t$ using CLIP's text encoder (Radford et al., 2021). We cap generation at 100 tokens, while CLIP accepts 77 tokens; longer descriptions are truncated during encoding. Although truncation discards some details, our method focuses on shared semantic cues (e.g., "short legs," "brown markings") rather than exhaustive captions, making it robust to this limitation. In parallel, we obtain the image embedding $z_x^v$ from the image encoder, and align the two embeddings with a contrastive loss:

$$\mathcal{L}_{\text{text}} = -\frac{1}{N} \sum_{i=1}^{N} \log \left( \frac{\exp(\text{sim}(z_i^v, z_i^t)/\tau)}{\sum_{j=1}^{N} \exp(\text{sim}(z_i^v, z_j^t)/\tau)} \right), \tag{1}$$

where $\text{sim}(\cdot, \cdot)$ is cosine similarity and $\tau$ is a temperature parameter. This loss guides the encoder to capture salient, label-independent traits shared across levels. Although not explicitly predicting attributes, aligning image features with text induces intermediate representations, which we call **pseudo-attributes**. This model-agnostic method can be applied to any architecture.

Finally, for hierarchical supervision, we apply the loss only at levels with available labels. Given hierarchical labels $y_1, \ldots, y_L$ across $L$ levels, the model computes a loss at each level:

$$\mathcal{L}_{\text{hier}} = \sum_{l=1}^{L} \mathbb{1}_{\{y_l \text{ exists}\}} \cdot \mathcal{L}(f_l(x), y_l), \tag{2}$$

where $f_l(x)$ is the prediction at level $l$, and $\mathcal{L}$ denotes any classification loss (e.g., cross-entropy).

**(2) Visual Guidance: Taxonomy-Guided Semi-Supervised Learning (Taxon-SSL).**

To enforce semantic consistency, we extend CHMatch's contrastive objective to the full taxonomy. For each mini-batch, we build level-wise affinity graphs $W^l$ based on pseudo-label agreement: $W_{ij}^l = 1$ if images $i$ and $j$ share the same pseudo-label at level $l$, and 0 otherwise. Then the taxonomy-aligned affinity graph $W$ is defined as:

$$W_{ij} = \begin{cases} 1 & \text{if } W_{ij}^1 = \ldots = W_{ij}^L = 1, \\ 0 & \text{otherwise.} \end{cases} \tag{3}$$

Then, taxonomy-aligned contrastive loss $\mathcal{L}_{\text{tacl}}$ is defined by:

$$\mathcal{L}_{\text{tacl}} = -\frac{1}{\sum_j W_{ij}} \cdot \sum_{l=1}^{L} \log \frac{\sum_j W_{ij} \exp((g(f(x_i)) \cdot g(f(x_j))')/t)}{\sum_j (1 - W_{ij}) \exp((g(f(x_i)) \cdot g(f(x_j))')/t)}, \tag{4}$$

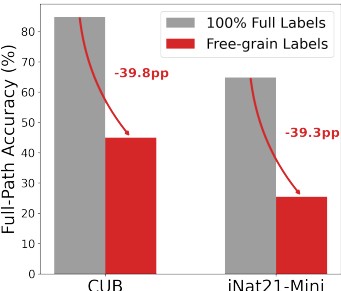 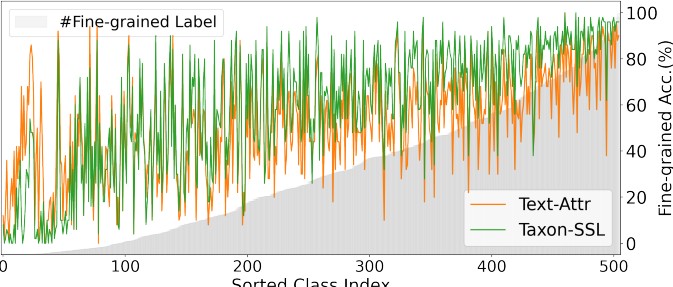

Figure 7: **Transitioning from fully labeled data to our mixed-granularity setting results in a substantial drop in Full-Path Accuracy, highlighting the difficulty of the task.** SOTA H-CAST suffers nearly a 40pp loss on both CUB and iNat21-mini.

Figure 8: **Text-Attr benefits under extreme label sparsity, as seen on the left (low-index classes with few fine-grained labels) by providing extra guidance from textual descriptions, while Taxon-SSL performs better on the right (high-index classes with more fine-grained labels).** Both are based on ViT-small model and evaluated on the ImageNet-Free. Classes are sorted by the number of fine-grained training samples, from lowest to highest.

where $g_i = g(f(x_i))$ is the projected feature of image $i$ with the classifier $f$, and $t$ is a temperature hyperparameter.

**(3) Combining Semantic and Visual Guidance: Taxon-SSL + Text-Attr.** A natural next step is to combine Text-Attr and Taxon-SSL by incorporating text-derived embeddings into the feature extractor of Taxon-SSL, allowing semantic and visual guidance to be jointly leveraged during training.

**(4) State-of-the-art Hierarchical Classification Methods: H-CAST, HRN.** We adopt two representative models. **(4-1) Hierarchical Residual Network (HRN)** (Chen et al., 2022): the first to handle supervision at both subordinate and fine-grained levels by maximizing marginal probabilities within the tree-constrained space. **(4-2) H-CAST** (Park et al., 2025): the current state-of-the-art, encouraging consistent visual grouding across taxonomy levels. Originally trained with full supervision, we adapt it to this setting via the level-wise loss in Eq. 2, using only available labels.

## 5 EXPERIMENTS

**Dataset:** We conduct experiments using our proposed **ImageNet-Free**, **iNat21-mini-Free**, and **CUB-Free** datasets, along with the synthetic **CUB-Rand** and **Aircraft-Rand** datasets. CUB includes bird images across 13 orders (e.g., *Anseriformes*), 38 families (e.g., *Anatidae*), and 200 species (e.g., *Mallard*), while Aircraft (Maji et al., 2013) contains aircraft images across 30 makers (e.g., *Boeing*), 70 families (e.g., *Boeing 707*), and 100 models (e.g., *707-320*).

**Evaluation metrics:** Following (Park et al., 2025), we evaluate accuracy and consistency: **1) Level-accuracy**: Top 1 accuracy for each level. **2) Tree-based InConsistency Error rate (TICE)**: Proportion of test samples with inconsistent prediction paths in the hierarchy. Lower is better. $\text{TICE} = \frac{n_{ic}}{N}$ **3) Full-Path Accuracy (FPA)**: Proportion of test samples with correct predictions at all hierarchy levels. Higher is better, and we use FPA as one of our primary metrics: $\text{FPA} = \frac{n_{ac}}{N}$.

**Implementation:** We use H-ViT, a ViT-Small-based hierarchical classifier, as the backbone for evaluating both Text-Attr and Taxon-SSL. To evaluate its compatibility across architectures, we also apply Text-Attr to H-CAST (Park et al., 2025), a state-of-the-art hierarchical model with comparable capacity. HRN (Chen et al., 2022) is evaluated with its original ResNet-50 backbone, which has over twice the parameters. All models are trained for 100 epochs, except for ImageNet-Free, which is trained for 200 due to its larger scale. Full architectural and training details are in the appendix F.

**Result 1: Performance Drop under Free-Grain Learning.** The prior hierarchical SOTA, H-CAST, degrades sharply under mixed-granularity labels on both CUB and iNat21-mini. As shown in Fig. 7, full-path accuracy drops from 84.9% to 45.1% on CUB-Free and from 64.9% to 25.6% on iNat21-mini-Free. This demonstrates the difficulty of handling mixed-granularity labels and imbalanced supervision across the hierarchy and need for methods handling them.

Table 3: **No single recipe solves free-grain learning—methods behave differently depending on data characteristics.** 1) Conventional hierarchical classification methods like HRN (Chen et al., 2022) and H-CAST (Park et al., 2025) show significant performance drops under incomplete supervision, underscoring the challenge of free-grain settings. 2) Text-Attr methods works well on ImageNet-Free, where each class is supported by abundant visual evidence. In contrast, iNat21-mini-Free has fine-grained biology labels, where appearance are similar, making LLM-based text descriptions less effective. Here, Taxon-SSL proves more beneficial by leveraging structured label propagation in this semi-supervised style setting. 3) Combining the two (Taxon-SSL + Text-Attr) yields consistent but modest gains across both datasets.

| Dataset | ImageNet-Free (20-127-505) | | | | | iNat21-mini-Free (273 - 1,103 - 10,000) | | | | |
|---|---|---|---|---|---|---|---|---|---|---|
| | FPA(↑) | fine.(↑) | sub.(↑) | basic(↑) | TICE(↓) | FPA(↑) | spec.(↑) | fam.(↑) | order(↑) | TICE(↓) |
| HRN (Chen et al., 2022) | 37.79 | 38.73 | 55.73 | 78.65 | 46.69 | 17.03 | 25.43 | 46.51 | 70.20 | 53.81 |
| H-CAST (Park et al., 2025) | 57.59 | 59.02 | 82.69 | 93.53 | 21.81 | 25.63 | 28.61 | 67.20 | 83.62 | 47.17 |
| Taxon-SSL | 48.40 | 52.34 | 65.74 | 82.96 | 19.87 | 31.74 | 37.11 | 69.53 | 82.02 | 37.31 |
| Taxon-SSL + Text-Attr | 49.65 | 53.43 | 66.43 | 83.56 | 18.81 | 31.93 | 37.08 | 69.76 | 82.20 | 37.04 |
| Text-Attr (H-ViT) | 55.48 | 59.05 | 77.95 | 89.45 | 24.02 | 27.88 | 32.07 | 68.27 | 80.49 | 46.35 |
| Text-Attr (H-CAST) | 63.20 | 64.91 | 84.47 | 93.56 | 18.58 | 29.74 | 32.37 | 71.79 | 85.99 | 44.63 |

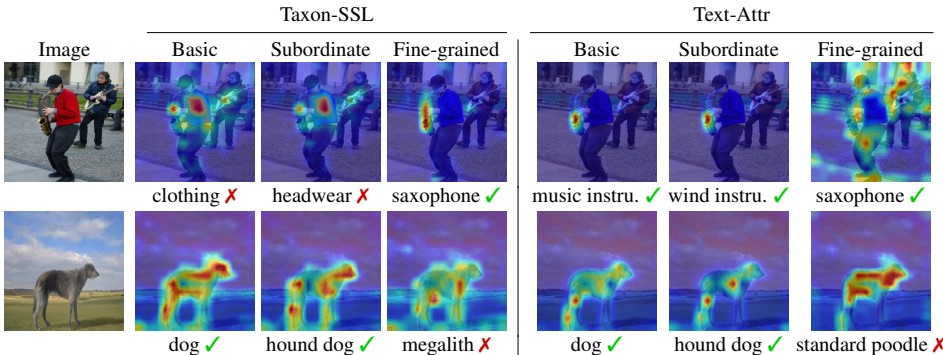

Figure 9: **Text-Attr improves semantic focus under diverse large-scale data.** (**1st row**) In a multi-object image, Taxon-SSL assigns inconsistent labels ("*clothing*" at the basic level, "*saxophone*" at the fine-grained level), while Text-Attr (H-ViT) correctly predicts "*musical instrument*" by focusing on the relevant object. (**2nd row**) When both fail at the fine-grained level, Taxon-SSL outputs an unrelated class ("*megalith*"), whereas Text-Attr (H-ViT) chooses a semantically closer one ("*poodle*"). This shows that text-derived attributes help the model attend to meaningful regions and maintain semantic plausibility, on large-scale ImageNet-Free dataset with diverse categories and sparse labels. Green/Red denote correct/incorrect predictions.

**Result 2: Performance on ImageNet-Free.** As shown in Table 3, existing hierarchical methods degrade sharply under free-grain learning: HRN reaches only 37.8% FPA, while H-CAST performs better at 57.6% but still struggles with missing labels. Text-Attr (H-ViT) achieves 55.5% without relying on H-CAST's visual grouping, and integrating it into H-CAST further improves performance to 63.2%, demonstrating the effectiveness of semantic-guided pseudo-attribute learning at scale. Taxon-SSL improves over HRN by leveraging visual guidance but remains less effective than Text-Attr methods, whose strong performance benefits from the abundance and diversity of ImageNet-Free for reliable visual–semantic alignment.

**Result 3: Performance on iNat21-mini-Free.** In Table 3, on the large-scale iNat21-mini-Free dataset, which contains many classes (10,000), conventional hierarchical methods perform poorly (17.0% for HRN, 25.63% for H-CAST). Taxon-SSL achieves the best performance (31.9% FPA), highlighting the benefits of structural label propagation under limited per-class supervision. Text-Attr methods perform slightly lower (27.9–30.0% FPA), likely due to restricted textual diversity in this fine-grained biological domain, yet still outperform conventional baselines.

In Appendix, we report additional results on CUB-Free (Sec. C.1), highly-missing *synthetic* datasets (Sec. C.2), and ablations on Text-Attr features, training strategies, and architecture design (Sec. E).

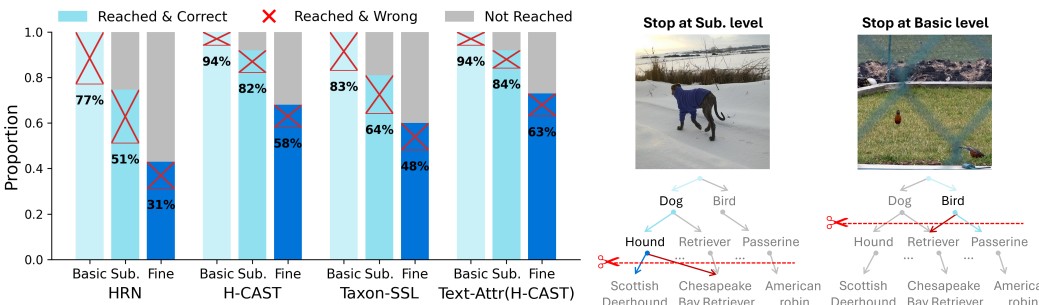

(a) **Consistency-based stopping for free-grain inference.** Predictions are halted when finer-level outputs conflict with preceding coarser-level predictions. On ImageNet-Free, Text-Attr (H-CAST) explores deeper levels of the hierarchy with higher correctness, whereas HRN stops earlier and produces fewer fine-level predictions.

(b) **Examples of consistency-based stopping in Text-Attr (H-CAST).** The model stops at the correct subordinate level (*Hound*, left) or at the basic level (*Bird*, right) when deeper predictions become inconsistent and incorrect, leading to more reliable results.

Figure 10: **Free-grain inference results with consistency-based stopping on ImageNet-Free.**

**Analysis 1: Text-Attr Excels with Sparse Labels, Taxon-SSL with Moderate Label Availability.** We analyze class-wise performance under imbalanced fine-grained label availability on ImageNet-Free. To isolate effects, we compare Text-Attr (H-ViT) and Taxon-SSL with identical ViT-small backbones, excluding H-CAST modules. Fig. 8 shows per-class accuracy, sorted by the number of fine-grained training labels. Text-Attr (H-ViT) outperforms in label-scarce classes by leveraging textual descriptions as extra supervision, while Taxon-SSL performs better with moderate label availability by propagating consistency across missing levels. We provide additional t-SNE (Maaten & Hinton, 2008) visualization analysis in Appendix D.

**Analysis 2: What Advantage Does External Semantic Guidance Provide?** To assess the effect of text-derived guidance, we compare saliency maps (Chefer et al., 2021) from Taxon-SSL and Text-Attr (H-ViT) (Fig. 9). In Row 1, with multiple objects, Taxon-SSL focuses on a human shoulder and misclassifies the image, violating the semantic hierarchy, while Text-Attr consistently attends to the instrument and predicts correctly. In Row 2, when both fail at the fine-grained level, Taxon-SSL outputs an unrelated class, whereas Text-Attr chooses a visually similar dog by focusing on curly fur and body shape. These results show that external semantic cues guide attention to meaningful features across label granularities, improving *hierarchical consistency*, while Taxon-SSL may drift to visually salient but semantically irrelevant regions under sparse or ambiguous supervision.

**Analysis 3: Free-grain Inference.** While our main goal is full-hierarchy prediction under mixed-granularity supervision, free-grain inference is also crucial in practice: a correct coarse label is often preferable to an incorrect fine-grained one (e.g., predicting "*dog*" instead of a wrong breed). We adopt a simple consistency-based stopping rule: predictions halt whenever the next-level label would violate the taxonomy, ensuring the deepest valid output. As shown in Fig. 10a, Text-Attr (H-CAST) reaches deeper levels more often and with higher accuracy. Figure 10b shows examples: stopping at the basic level when "*bird*" is correct but the subordinate mispredicts, or at the subordinate level when "*dog → hound*" is correct but the fine-grained label is inconsistent. These results highlight the practical value of free-grain inference and motivate benchmarks that explicitly evaluate this setting.

## 6 Summary

We introduce new hierarchical classification under free-grain supervision, where models learn from labels of varying granularity while maintaining taxonomy consistency. To advance this setting, we present a large-scale benchmark and two simple yet effective baselines. Our Text-Attr method mitigates label imbalance by sharing features across levels, though it does not explicitly model it; future work could explore imbalance-aware strategies for further improvement.

**Ethics statement.** This work aims to improve learning under coarse or incomplete supervision, supporting applications in low-resource settings where fine-grained annotation is hard to obtain.

**Reproducibility statement.** Appendix F provides implementation details. Training code and dataset for ImageNet-Free is included in the supplementary material, and the full code will be released publicly upon acceptance.

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

# Free-Grained Hierarchical Recognition

## Supplementary Material

CONTENTS

# A  COMPLETE HIERARCHY OF IMAGENET-FREE

| Basic | Subordinate | Fine-Grained |
|---|---|---|
| bird | passerine bird | brambling, indigo bunting, robin, jay, bulbul, water ouzel, house finch, chickadee, junco, magpie, goldfinch |
| | parrot | macaw, sulphur-crested cockatoo, African grey, lorikeet |
| | piciform bird | toucan, jacamar |
| | seabird | king penguin, pelican, albatross |
| | anseriform bird | drake, red-breasted merganser, black swan, goose |
| | coraciiform bird | bee eater, hornbill |
| | bird of prey | kite, great grey owl, vulture, bald eagle |
| | gallinaceous bird | partridge, prairie chicken, ruffed grouse, peacock, quail, black grouse, ptarmigan |
| | wading bird | flamingo, American coot, redshank, American egret, little blue heron, white stork, limpkin, spoonbill, red-backed sandpiper, dowitcher, crane, ruddy turnstone, bittern, oystercatcher, black stork, bustard |
| dog | spitz dog | malamute, Pomeranian, keeshond, Siberian husky, chow, Samoyed |
| | pointer dog | vizsla, German short-haired pointer |
| | spaniel dog | Brittany spaniel, clumber, English springer, Sussex spaniel, Irish water spaniel, Welsh springer spaniel, cocker spaniel |
| | hound dog | basset, bloodhound, Irish wolfhound, Walker hound, redbone, English foxhound, Italian greyhound, Ibizan hound, bluetick, Scottish deerhound, borzoi, Norwegian elkhound, whippet, Weimaraner, Saluki, beagle, Afghan hound, black-and-tan coonhound, otterhound |
| | terrier dog | Boston bull, silky terrier, Lakeland terrier, Yorkshire terrier, Tibetan terrier, American Staffordshire terrier, Irish terrier, Airedale, Norwich terrier, soft-coated wheaten terrier, wire-haired fox terrier, Staffordshire bullterrier, West Highland white terrier, Australian terrier, Dandie Dinmont, Kerry blue terrier, Lhasa, cairn, Sealyham terrier, Bedlington terrier, Scotch terrier, Border terrier, Norfolk terrier |
| | corgi dog | Pembroke, Cardigan |
| | poodle dog | miniature poodle, toy poodle, standard poodle |
| | setter dog | Irish setter, Gordon setter, English setter |
| | pinscher dog | Doberman, affenpinscher, miniature pinscher |

|  |  | shepherd dog | kelpie, briard, German shepherd, Old English sheepdog, Border collie, Bouvier des Flandres, collie, Rottweiler, komondor, malinois, groenendael, Shetland sheepdog |
|  |  | retriever dog | curly-coated retriever, Labrador retriever, Chesapeake Bay retriever, flat-coated retriever, golden retriever |
|  |  | schnauzer dog | standard schnauzer, miniature schnauzer, giant schnauzer |
|  |  | Sennenhunde dog | Bernese mountain dog, Greater Swiss Mountain dog, Appenzeller, EntleBucher |
|  |  | toy dog | toy terrier, Blenheim spaniel, Maltese dog, Shih-Tzu, papillon, Pekinese, Chihuahua, Japanese spaniel |
| fish |  | soft-finned fish | coho, tench, eel, goldfish |
|  |  | shark | tiger shark, great white shark, hammerhead |
|  |  | spiny-finned fish | anemone fish, puffer, lionfish, rock beauty |
|  |  | ray | stingray, electric ray |
|  |  | ganoid fish | sturgeon, gar |
| primate |  | ape | gibbon, siamang, orangutan, chimpanzee, gorilla |
|  |  | monkey | titi, langur, colobus, squirrel monkey, baboon, guenon, marmoset, macaque, spider monkey, patas, howler monkey, proboscis monkey, capuchin |
|  |  | lemur | Madagascar cat, indri |
| snake |  | colubrid snake | water snake, garter snake, green snake, night snake, hognose snake, ringneck snake, king snake, thunder snake, vine snake |
|  |  | elapid snake | sea snake, Indian cobra, green mamba |
|  |  | viper | diamondback, horned viper, sidewinder |
|  |  | boa snake | boa constrictor, rock python |
| salamander |  | newt | eft, common newt |
|  |  | ambystomid salamander | spotted salamander, axolotl |
| insect |  | beetle | dung beetle, weevil, leaf beetle, tiger beetle, ladybug, rhinoceros beetle, longhorned beetle, ground beetle |
|  |  | orthopterous insect | cricket, grasshopper |
|  |  | dictyopterous insect | cockroach, mantis |
|  |  | hymenopterous insect | bee, ant |
|  |  | butterflyinsect | cabbage butterfly, lycaenid, monarch, admiral, sulphur butterfly, ringlet |
|  |  | odonate insect | dragonfly, damselfly |
|  |  | homopterous insect | cicada, leafhopper |
| furniture |  | table | desk, dining table |
|  |  | baby bed | cradle, crib, bassinet |
|  |  | seat | rocking chair, barber chair, park bench, throne, folding chair, toilet seat, studio couch |

| | lamp | table lamp |
| --- | --- | --- |
| | cabinet | china cabinet, medicine chest |
| musical instrument | wind instrument | ocarina, flute, panpipe, oboe, cornet, sax, harmonica, bassoon, French horn, trombone |
| | stringed instrument | banjo, harp, violin, cello, acoustic guitar, electric guitar |
| | percussion instrument | steel drum, gong, marimba, drum, chime, maraca |
| | keyboard instrument | upright, grand piano, accordion, organ |
| scientific instrument | laboratory glassware | Petri dish |
| | magnifier | loupe, radio telescope |
| sports equipment | ball | golf ball, baseball, basketball, croquet ball |
| | gymnastic apparatus | parallel bars, balance beam, horizontal bar |
| | weight | barbell, dumbbell |
| electronic equipment | telephone | dial telephone, pay-phone, cellular telephone |
| | computer peripheral | printer, joystick, computer keyboard, mouse |
| | audio device | tape player, cassette player, CD player, iPod |
| | network device | modem |
| | display device | monitor, screen |
| clothing | bottoms (skirts) | hoopskirt, sarong, miniskirt, overskirt |
| | tops (sweaters) | sweatshirt, cardigan |
| | outwear | trench coat, poncho, fur coat |
| | swimwear | maillot, bikini, swimming trunks |
| | face & headwear | wig, sombrero, mortarboard, bonnet, mask, cowboy hat, bearskin |
| | nightwear | pajama |
| | protective wear | apron, knee pad, lab coat |
| | dresses & Gowns | gown |
| | underwear | brassiere |
| | footwear | sock, Christmas stocking |
| | neckwear | bow tie, bolo tie, Windsor tie |
| | traditional & formal Wear | abaya, kimono, vestment, academic gown |
| | wraps & shawls | stole, feather boa |
| container | reservoir | water tower, rain barrel |
| | bag | mailbag, plastic bag, backpack, purse |
| | jug | water jug, whiskey jug |
| | vessel | mortar, pitcher, tub, ladle, bucket, coffee mug |
| | bottle | wine bottle, beer bottle, pop bottle, water bottle, pill bottle |
| | basket | hamper, shopping basket |
| | box | mailbox, carton, pencil box, chest, crate |

| | glass | goblet, beer glass |
|---|---|---|
| | shaker | saltshaker, cocktail shaker |
| cooking utensil | pan | frying pan, wok |
| | cooker | Crock Pot |
| | pot | teapot, caldron, coffeepot |
| structure | monument | brass, megalith, triumphal arch, obelisk, totem pole |
| | religious building | church, mosque, boathouse, monastery, stupa |
| | housing | yurt, cliff dwelling, mobile home |
| | public building | planetarium, library |
| | movable structure | sliding door, turnstile |
| | supporting structure | plate rack, honeycomb, pedestal |
| | fence | stone wall, picket fence, chainlink fence, worm fence |
| | bridge | steel arch bridge, viaduct, suspension bridge |
| | residential structure | palace |
| | agricultural structure | greenhouse, barn, apiary |
| | commercial stucture | toyshop, restaurant, cinema, confectionery, bookshop, grocery store, tobacco shop, bakery, butcher shop, barbershop, shoe shop |
| | barrier | grille, bannister, breakwater, dam |
| | institutional structure | prison |
| tool | hand tool | hammer, plunger, screwdriver |
| | garden tool | lawn mower, shovel |
| | cutter | cleaver, plane, letter opener, hatchet |
| | power tool | chain saw |
| | opener | corkscrew, can opener |
| craft | sailing vessel | trimaran, schooner, catamaran |
| | boat | fireboat, canoe, yawl, gondola, speedboat, lifeboat |
| | ship | wreck, pirate, container ship, liner |
| | warship | aircraft carrier, submarine |
| | aircraft | airliner, warplane, airship, balloon |
| vehicle | bicycle | bicycle-built-for-two, mountain bike |
| | bus | minibus, school bus, trolleybus |
| | car | ambulance, beach wagon, cab, convertible, jeep, limousine, Model T, racer, sports car |
| | truck | fire engine, garbage truck, pickup, tow truck, trailer truck |
| | van | minivan, moving van, police van |
| | locomotive | electric locomotive, steam locomotive |
| | military vehicle | half track |

| | self-propelled vehicle | forklift, recreational vehicle, snowmobile, tank, tractor, golfcart, snowplow, go-kart, moped, streetcar, amphibious vehicle |
|---|---|---|
| | handcart | barrow, shopping cart |
| | sled | bobsled, dogsled |
| | train | bullet train |
| | wagon | horse cart, jinrikisha, oxcart |
| | wheeled vehicle | freight car, motor scooter, tricycle, unicycle |
| weapon | gun | rifle, assault rifle, revolver, cannon |
| | ranged weapon | missile, projectile |

Table 4: **Complete hierarchy tree for our proposed ImageNet-Free dataset.**

# B RELATED WORK

**Hierarchical classification** has been studied with varying objectives. Most focus on *leaf-node prediction*, using the full taxonomy during training but predicting only fine-grained labels (Karthik et al., 2021; Zhang et al., 2022; Zeng et al., 2022; Garg et al., 2022b). Evaluation in these works typically relies on top-1 accuracy or mistake severity at the leaf level, making them compatible with large-scale datasets like ImageNet (Russakovsky et al., 2015) and tieredImageNet (Ren et al., 2018)—even with inconsistent or deep hierarchies. However, models restricted to fine-grained outputs often fail in real-world scenarios where visual details are missing, as they cannot fall back to coarser labels and thus provide no meaningful information.

To address this, full taxonomy prediction has been explored, aiming to produce labels across all levels while maintaining hierarchical consistency (Chang et al., 2021; Wang et al., 2023; Jiang et al., 2024; Park et al., 2025). However, these methods are typically developed and evaluated on small, fully labeled datasets like CUB (Welinder et al., 2010) and Aircraft (Maji et al., 2013), which lack the scale, diversity, and label sparsity of real-world settings. The iNaturalist dataset (Van Horn et al., 2021) offers a deeper taxonomy, but also remains restricted to the biology, limiting its suitability for general-purpose evaluation. HRN (Chen et al., 2022) partially handles incomplete labels by randomly converting fine-grained labels to parent categories, overlooking the structured ambiguity in real data. Similarly, (Kim et al., 2023) supports mixed labels but treats them flatly, ignoring hierarchical relationships. Both also rely on small datasets such as CUB and Aircraft. Our work fills this gap by enabling full taxonomy prediction under realistic supervision on large-scale data.

**Imbalanced classification** has been extensively studied (Liu et al., 2019; Ren et al., 2020; Wang et al., 2021; Park et al., 2021; Tian et al., 2022; Park et al., 2022; Ha et al., 2023; Zhao et al., 2024), mostly focusing on intra-level imbalance at a single fine-grained level. In contrast, we address intra- and inter-level imbalance in a hierarchical setting, where classes are balanced but label granularity varies across them. DeepRTC (Wu et al., 2020) considers taxonomy, but aims to improve inference reliability via early stopping, rather than predicting the full taxonomy.

**Semi-supervised learning** typically combines labeled and unlabeled data at a single fine-grained level (Tarvainen & Valpola, 2017; Berthelot et al., 2019; Sohn et al., 2020). Recent work incorporates coarse labels (Garg et al., 2022a; Wu et al., 2023), but still targets fine-grained accuracy. In contrast, our setting demands consistent prediction across the full taxonomy with heterogeneous supervision, making existing methods not directly applicable.

**Weakly-supervised classification** typically aims to predict fine-grained labels when only coarse labels are available during training (Robinson et al., 2020; Grcic et al., 2024). These methods assume fully observed labels at a coarse level and focus on improving predictions at a fine-grained level. In contrast, our setting requires handling multi-granularity labels and inferring the full taxonomy.

**Foundation models for zero-shot classification**, such as vision-language models (e.g., CLIP (Radford et al., 2021)) and large language models (e.g., GPT-4 (Achiam et al., 2023)), have gained popularity for leveraging label-driven prompts at inference—without training (Pratt et al., 2023; Liu et al., 2024; Zheng et al., 2024; Saha et al., 2024). These methods aim to improve flat-level classification by matching images to text. In contrast, we train a hierarchical classifier that learns shared visual patterns across levels from images, when labels are partially missing. Our model requires no textual input at inference, making it efficient. See a full task comparison in Table 2.

# C  MORE EXPERIMANTAL RESULTS

## C.1  EVALUAION ON CUB-FREE

On the small-scale, single-domain dataset CUB-Free (Table 5), Taxon-SSL achieves the best performance (63.96% FPA), showing the advantage of structured label propagation when per-class samples are scarce. Text-Attr methods perform moderately well (53.99–57.59% FPA) but are less effective here, as the bird-only domain limits textual diversity and reduces the benefit of language-based supervision. Still, they clearly outperform conventional hierarchical baselines (44.30% for HRN, 45.10% for H-CAST), underscoring the overall effectiveness of our approach. Unlike the trend on large-scale, diverse datasets such as ImageNet-Free, where Text-Attr provides richer cues and stronger gains, these results confirm that there is no single recipe for free-grain learning: performance is tightly coupled with dataset characteristics, making the problem inherently challenging.

Table 5: **Taxon-SSL shows strong effectiveness on the small-scale dataset CUB-Free, where label propagation provides reliable supervision despite limited data.** Text-Attr methods are assumed to offer limited benefit due to the restricted textual diversity of this bird-only dataset.

| CUB-Free (13-38-200) | FPA (↑) | Species (↑) | family (↑) | Order (↑) | TICE (↓) |
|---|---|---|---|---|---|
| HRN (Chen et al., 2022) | 44.30 | 46.72 | 81.20 | 96.36 | 27.15 |
| H-CAST (Park et al., 2025) | 45.10 | 47.52 | 87.78 | 97.50 | 25.89 |
| Taxon-SSL | **63.96** | **65.50** | **92.84** | 98.40 | **7.39** |
| Taxon-SSL + Text-Attr | 63.05 | 64.86 | 92.54 | 98.38 | 7.61 |
| Text-Attr (H-ViT) | 57.59 | 59.10 | 91.60 | 98.05 | 10.72 |
| Text-Attr (H-CAST) | 53.99 | 55.58 | 91.72 | **98.41** | 18.95 |

## C.2  EVALUATION UNDER VARYING AND SEVERE LABEL SPARSITY CONDITIONS

To evaluate model performance under diverse and more challenging free-grain conditions, we experiment with various label availability ratios by randomly removing fine-grained labels—e.g., (100%-60%-30%), (100%-50%-10%), and (100%-20%-10%)—which represent the available proportions of basic, subordinate, and fine-grained labels, respectively. Each experiment is repeated with three different random seeds, and we report the average performance. The variance across runs was minor (0.1–1.8).

Consistent with our main results, these experiments (Table 6 & 7 & 8) also show that **there is *no single method that performs best across all settings***. Instead, the most effective method varies depending on the dataset and the specific ratio of available labels, highlighting the importance of adaptable free-grain learning strategies.

For consistency, we refer to the three levels in CUB-Rand (order-family-species) and Aircraft-Rand (maker-family-model) as basic, subordinate, and fine-grained levels. We summarize the key findings below:

**(1) Conventional hierarchical classification methods struggle under the free-grain setting, where label supervision is sparse and uneven across levels.** For example, when labels are highly missing (e.g., only 10% available at the fine-grained level), HRN (Chen et al., 2022) and H-CAST (Park et al., 2025) suffer more than a 50% drop in accuracy across all levels compared to the fully labeled (100%-100%-100%) setting on CUB-Rand (Fig. 6 & Table 8). This highlights the difficulty of the free-grain setting and the need for methods that can robustly handle incomplete supervision at multiple semantic levels.

**(2) The performance of different methods varies with the amount of available supervision per class:** Text-Attr methods perform better when more labeled samples are available, while Taxon-SSL is more effective under extreme label sparsity. For example, in Table 6, the average number of available fine-grained labels per class is approximately 9 for CUB-Rand and about 20 for Aircraft-Rand. Consistent with this difference, Taxon-SSL outperforms other methods on CUB-Rand, whereas Text-Attr (H-CAST) performs best on Aircraft-Rand. This trend persists across settings. In the most sparse setting, CUB-Rand (100-20-10, Table 8), where only about 3 fine-grained labels are available per class, Taxon-SSL shows a clear advantage. We attribute this to how supervision is utilized. Text-Attr relies on available labels and indirect semantic guidance via text features.

In contrast, Taxon-SSL actively leverages unlabeled data through pseudo-labeling and strong augmentations, making it more effective when labeled examples are extremely limited.

**(3) Sometimes, Taxon-SSL's high fine-grained accuracy comes at the cost of lower accuracy at higher levels in the taxonomy.** For example, in Table 7, Taxon-SSL achieves the highest fine-grained accuracy (65.01%), but its subordinate and basic-level accuracies (85.53% and 92.81%) are lower than those of Text-Attr (H-CAST), which achieves 86.30% and 94.17%, respectively. This highlights a key challenge in free-grain learning: improving accuracy across all levels simultaneously is non-trivial, and optimizing for fine-grained performance alone may degrade consistency at coarser levels.

Table 6: **No single method performs best across all conditions—performance depends strongly on the amount of available supervision per class.** Text-Attr methods tend to perform better when more labeled samples are available, while Taxon-SSL is more effective under extreme label sparsity. For example, Taxon-SSL performs best on CUB-Rand with around 9 fine-grained labels per class, while Text-Attr (H-CAST) performs best on Aircraft-Rand with around 20, reflecting the impact of supervision density. These results highlight that method effectiveness is highly sensitive to label sparsity, emphasizing the need for adaptable approaches in free-grain learning.

| Label Ratio | CUB-Rand (100%-60%-30%) | | | | | Aircraft-Rand (100%-60%-30%) | | | | |
|---|---|---|---|---|---|---|---|---|---|---|
| | FPA(↑) | spec.(↑) | fam.(↑) | order(↑) | TICE(↓) | FPA(↑) | maker(↑) | fam.(↑) | model(↑) | TICE(↓) |
| HRN (Chen et al., 2022) | 57.87 | 62.73 | 85.53 | 96.45 | 13.77 | 57.33 | 64.42 | 76.95 | 86.38 | 23.30 |
| H-CAST (Park et al., 2025) | 61.88 | 67.36 | 90.05 | 94.32 | 13.04 | 64.67 | 68.88 | 85.58 | 91.43 | 13.76 |
| Taxon-SSL | 74.82 | 76.92 | 93.38 | 98.33 | 5.06 | 70.33 | 72.22 | 87.06 | 93.50 | **7.18** |
| Taxon-SSL + Text-Attr | **74.90** | **76.95** | **93.41** | 98.38 | **4.91** | 69.89 | 72.24 | 86.92 | 93.29 | 7.77 |
| Text-Attr (H-ViT) | 67.89 | 72.48 | 90.63 | 95.37 | 10.39 | 64.15 | 68.92 | 85.88 | 89.87 | 15.80 |
| Text-Attr (H-CAST) | 69.65 | 71.31 | 92.88 | **98.48** | 8.35 | **71.43** | **73.56** | **89.66** | **95.31** | 9.71 |

Table 7: **Maintaining accuracy across all hierarchy levels remains more challenging under sparse supervision.** For example, in 100%-50%-10% case, Taxon-SSL achieves the highest fine-grained accuracy (65.01%), but its subordinate and basic-level accuracies (85.53%, 92.81%) are lower than those of Text-Attr (H-CAST) (86.30%, 94.17%), which better preserves consistency across levels. This result illustrates the inherent difficulty of improving accuracy across all levels simultaneously, as objectives at different levels can be conflicting.

| Label Ratio | Aircraft-Rand (100%-50%-10%) | | | | | Aircraft-Rand (100%-20%-10%) | | | | |
|---|---|---|---|---|---|---|---|---|---|---|
| | FPA(↑) | maker(↑) | fam.(↑) | model(↑) | TICE(↓) | FPA(↑) | maker(↑) | fam.(↑) | model(↑) | TICE(↓) |
| HRN (Chen et al., 2022) | 40.35 | 47.85 | 70.76 | 85.68 | 37.56 | 32.06 | 46.73 | 55.43 | 85.58 | 48.43 |
| H-CAST (Park et al., 2025) | 47.57 | 51.93 | 78.31 | 87.11 | 28.42 | 40.33 | 45.44 | 67.28 | 84.12 | 35.61 |
| Taxon-SSL | 62.61 | 65.01 | 85.53 | 92.81 | **10.22** | **58.73** | **61.10** | 80.90 | 92.24 | **11.77** |
| Taxon-SSL + Text-Attr | **62.95** | **65.49** | 86.01 | 92.64 | 10.25 | 58.55 | 60.88 | **80.97** | 92.04 | 11.89 |
| Text-Attr (H-ViT) | 47.83 | 52.25 | 81.13 | 87.82 | 30.57 | 38.73 | 43.89 | 66.13 | 84.81 | 38.69 |
| Text-Attr (H-CAST) | 53.31 | 55.32 | **86.30** | **94.17** | 24.43 | 48.85 | 51.37 | 77.11 | **93.01** | 27.25 |

Table 8: **Taxon-SSL is more robust under extreme label sparsity, while other methods degrade significantly.** In CUB-Rand (100%-20%-10%), where each class has only 3 fine-grained and 3 subordinate labels, Taxon-SSL achieves the best performance, while other methods struggle. HRN and H-CAST suffer over 50% drop in fine-grained accuracy compared to the fully-supervised (100%-100%-100%) setting. Text-Attr methods perform more robustly (10%+ higher than HRN/H-CAST), but still fall short under such sparse supervision. We attribute this to how each method leverages supervision: Text-Attr relies on available labels and semantic guidance from text features, while Taxon-SSL benefits more from unlabeled data via pseudo-labeling and augmentations, making it more effective under severe label sparsity.

| Label Ratio | CUB-Rand (100%-50%-10%) | | | | | CUB-Rand (100%-20%-10%) | | | | |
|---|---|---|---|---|---|---|---|---|---|---|
| | FPA(↑) | spec.(↑) | fam.(↑) | order(↑) | TICE(↓) | FPA(↑) | spec.(↑) | fam.(↑) | order(↑) | TICE(↓) |
| HRN (Chen et al., 2022) | 40.23 | 43.70 | 82.75 | 95.94 | 22.34 | 33.53 | 41.18 | 72.56 | 95.79 | 30.50 |
| H-CAST (Park et al., 2025) | 39.03 | 43.41 | 85.74 | 93.23 | 24.60 | 32.97 | 38.66 | 76.89 | 92.50 | 29.43 |
| Taxon-SSL | 62.40 | 64.14 | **92.33** | **98.26** | **6.01** | **59.18** | **61.44** | **89.79** | **98.20** | **7.65** |
| Taxon-SSL + Text-Attr | **62.52** | **64.87** | 87.94 | 94.45 | 8.98 | 57.98 | 60.59 | 89.42 | 98.12 | 8.39 |
| Text-Attr (H-ViT) | 47.42 | 50.74 | 88.22 | 94.67 | 18.09 | 42.46 | 46.99 | 80.92 | 94.43 | 20.27 |
| Text-Attr (H-CAST) | 44.63 | 45.89 | 91.06 | 98.19 | 22.72 | 40.41 | 42.76 | 84.24 | 97.97 | 24.05 |

# D  T-SNE VISUALIZATION

We visualize ImageNet-Free embeddings of Text-Attr (H-CAST) and Taxon-SSL using t-SNE (Maaten & Hinton, 2008) to assess whether the learned representations capture semantic and hierarchical structure. Each point denotes an image embedding, colored by its basic-level class (20 categories), with brightness variations indicating fine-grained subclasses (505 total).

Both Text-Attr (H-CAST) and Taxon-SSL produce well-separated clusters consistent with the basic-level taxonomy, showing that coarse groupings are reliably captured. The key difference lies within coarse categories: **Text-Attr (H-CAST) reveals more distinct fine-grained subclusters** (e.g., breeds within *dog*, species within *bird*), whereas **Taxon-SSL yields tighter coarse clusters with less apparent fine-level separation**.

This contrast reflects their supervision signals. Text-Attr leverages diverse textual cues (attributes, parts, appearance terms), which promote discriminative, attribute-aligned features and sharpen within-class distinctions. Taxon-SSL, by propagating labels along the taxonomy and enforcing consistency under mixed-granularity supervision, regularizes embeddings within each coarse class and reduces intra-class variance—emphasizing coarse alignment over fine-level separability.

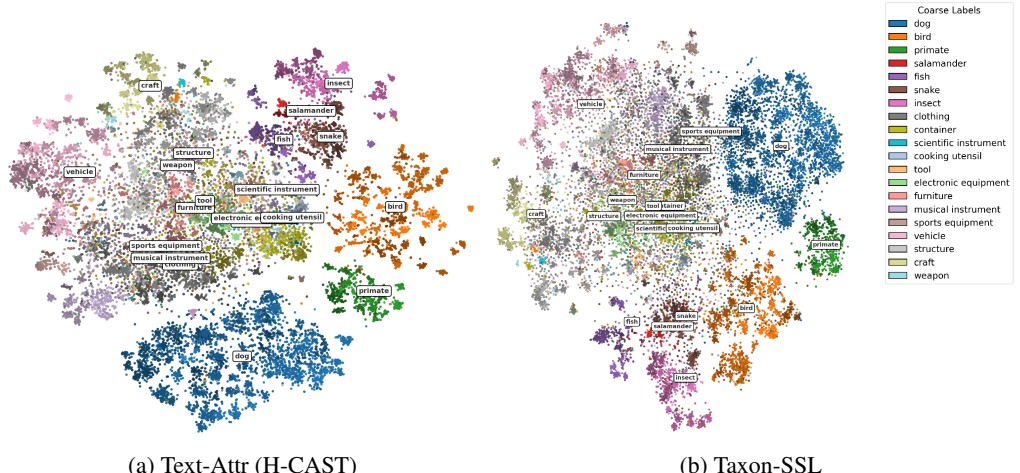

(a) Text-Attr (H-CAST)           (b) Taxon-SSL

Figure 11: **t-sne Visualization on ImageNet-Free.** Both methods separate coarse-level taxonomy well, but Text-Attr (H-CAST) yields clearer fine-grained subclusters (e.g., distinct groups within *dog* and *bird*) with more compact grouping, whereas Taxon-SSL shows some overlap of embeddings near cluster boundaries. This is likely due to ImageNet-Free 's diverse large-scale categories, where text supervision provides rich attribute cues that sharpen fine-level distinctions.

# E ABLATION STUDY

## E.1 IMPORTANCE OF TEXT-GUIDED PSEUDO ATTRIBUTES

Text-guided Pseudo Attributes jointly optimizes hierarchical label supervision ($\mathcal{L}_{\text{hier}}$) and text-guided pseudo attributes ($\mathcal{L}_{\text{text}}$) to learn semantically rich features: $\mathcal{L} = \mathcal{L}_{\text{hier}} + \alpha\mathcal{L}_{\text{text}}$ Fig. 12 quantifies $\mathcal{L}_{\text{text}}$'s impact by varying its weight $\alpha$ on CUB-Rand. Ablating $\mathcal{L}_{\text{text}}$ ($\alpha = 0$) causes a 5% absolute decline in both fine-grained accuracy and FPA compared to the optimal configuration ($\alpha = 0$). This gap underscores two key roles of text guidance: (1) it injects complementary visual semantics absent in class labels alone, and (2) it enforces attribute consistency across hierarchy levels. The performance recovery at ($\alpha = 1$) confirms that textual pseudo-attributes mitigate annotation sparsity while preserving taxonomic coherence.

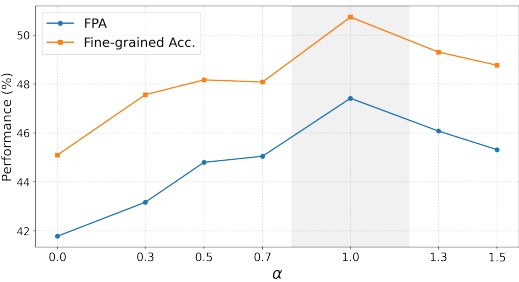

Figure 12: **Tuning $\alpha$ balances accuracy and taxonomic consistency.** At $\alpha = 1$ (optimal), Text-Attr (H-ViT) achieves peak fine-grained accuracy (blue) while maintaining hierarchical consistency (orange). Ablating $\mathcal{L}_{\text{text}}$ ($\alpha = 0$) causes a 5% accuracy drop and increased inconsistency, as class embeddings lose text-guided attribute alignment. Higher $\alpha > 1.0$ over-regularizes features, marginally degrading both metrics. This trade-off underscores the need to weight text supervision to resolve sparse annotations without distorting the hierarchy.

## E.2 COMBINING TEXT-ATTR AND TAXON-SSL

We compare different training schedules for combining Text-Attr and Taxon-SSL on CUB-Free. In the **joint setting**, both objectives are optimized simultaneously for 100 epochs. In the **two-stage setting**, we first train with one objective for 50 epochs and then add the other for the remaining 50 epochs, considering both orders: (1) Taxon-SSL $\rightarrow$ Text-Attr, and (2) Text-Attr $\rightarrow$ Taxon-SSL.

Table 9 show that starting with Text-Attr and then adding Taxon-SSL yields slightly higher full-path accuracy, likely because textual supervision promotes diverse feature learning before label propagation. In contrast, beginning with Taxon-SSL provides no advantage, and both two-stage variants perform similarly to joint training overall. Interestingly, joint training achieves higher consistency as measured by TICE. Given its simplicity and competitive performance, we adopt the joint strategy as our default.

Table 9: **Comparison of joint vs. two-stage training schedules for Text-Attr and Taxon-SSL on CUB-Free.** While two-stage training (Text-Attr $\rightarrow$ Taxon-SSL) yields slightly higher accuracy, joint learning is simpler and provides better consistency (TICE).

| CUB-Free (13-38-200) | FPA ($\uparrow$) | Species ($\uparrow$) | family ($\uparrow$) | Order ($\uparrow$) | TICE ($\downarrow$) |
|---|---|---|---|---|---|
| Taxon-SSL + Text-Attr (100 epochs) | 63.04 | 64.86 | 92.54 | **98.37** | **7.61** |
| Taxon-SSL (50 epochs) $\rightarrow$ +Text-Attr (50 epochs) | 62.84 | 64.42 | 92.47 | 98.20 | 8.19 |
| Text-Attr (50 epochs) $\rightarrow$ +Taxon-SSL (50 epochs) | **63.63** | **65.34** | **92.56** | 98.27 | 8.06 |

## E.3 ABLATION ON HIERARCHICAL SUPERVISION IN VIT

We further examine the architectural design choice of where to inject hierarchical supervision in the Vision Transformer (ViT) in Table 10. On CUB-Free, we map the three taxonomy levels (Or-

der–Family–Species) to different layers and compare multiple configurations: (6th, 9th, 12th), (8th, 10th, 12th), and (10th, 11th, 12th).

Among these, supervision at the 8th, 10th, and 12th layers yields the best performance. We interpret this as a balance between early and late representation learning: assigning hierarchy too early (e.g., 6–9–12) forces the model to align coarse categories before sufficient visual features are developed, while placing all supervision too late (e.g., 10–11–12) limits the model's capacity to gradually refine class granularity. The 8–10–12 configuration provides an appropriate middle ground, where lower-level categories benefit from moderately abstract features, and finer distinctions are introduced after the backbone has matured.

Table 10: **Performance comparison of different layer assignments for hierarchical supervision in ViT on CUB-Free.** The 8th–10th–12th configuration achieves the best results, balancing early and late feature abstraction.

| CUB-Free (13-38-200) | FPA ($\uparrow$) | Species ($\uparrow$) | family ($\uparrow$) | Order ($\uparrow$) | TICE ($\downarrow$) |
|---|---|---|---|---|---|
| 6-9-12th layer | 54.80 | 58.16 | 88.97 | 95.01 | 16.79 |
| 8-10-12th layer | **57.59** | **59.10** | **91.60** | **98.05** | **10.72** |
| 10-11-12th layer | 56.40 | 58.56 | 90.80 | 97.08 | 13.48 |

## F    IMPLEMENTATION DETAILS

For ViT (Dosovitskiy et al., 2020) models, we use ViT-Small for Text-Attr (H-ViT) and Taxon-SSL and H-CAST-Small (Park et al., 2025) for Text-Attr (H-CAST) to match parameter sizes.

For Text-Attr (H-ViT), we insert fully-connected layers to the class token at the 8th, 10th, and 12th layers for basic, subordinate, and fine-grained supervision. The 12th-layer patch features are projected to match the text embedding dimension via an FC layer. For Text-Attr (H-CAST), hierarchical supervision is applied to the last three blocks, following (Park et al., 2025). Due to low dimensionality in the final block, we align text features with the features of the second block. For Text-Attr methods, CLIP-ViT-B/32 is used to extract text embeddings, which remain frozen during training.

In Taxon-SSL, we apply a shared MLP to the class token from the final (12th) layer, followed by three separate linear classifiers for basic, subordinate, and fine-grained supervision. When combined with Text-Attr, we additionally project the class token through a linear layer and align it with the corresponding text feature.

For hierarchical classification baselines, HRN (Chen et al., 2022) and H-CAST (Park et al., 2025), we follow their original training protocols and retrain them under our free-grain setting. We extend HRN to handle missing labels at two levels instead of one. For H-CAST, we provide supervision using the available labels at each corresponding level. Full hyperparameter configurations are provided in Table 11.

We train all models for 100 epochs, except for ImageNet-Free, which are trained for 200 epochs due to the larger scale. All experiments were conducted on an NVIDIA A40 GPU with 48GB memory. We used a single GPU for all experiments, except for ImageNet-Free, which was trained using 4 GPUs. We include our training code for ImageNet-Free in the supplementary material and will release the full code publicly upon acceptance.

Table 11: **Hyperparameters for training Text-Attr (H-ViT), Text-Attr (H-CAST), and Taxon-SSL.** We follow the training setup of H-CAST (Park et al., 2025) for Text-Attr methods (Text-Attr (H-ViT) and Text-Attr (H-CAST)), and adopt the settings of CHMatch (Wu et al., 2023) for Taxon-SSL.

| Parameter | Text-Attr (H-ViT) | Text-Attr (H-CAST) | Taxon-SSL |
|---|---|---|---|
| batch_size | 256 | 256 | 128 |
| crop_size | 224 | 224 | 224 |
| learning_rate | $5e-4$ | $5e-4$ | $1e-3$ |
| weight_decay | 0.05 | 0.05 | 0.05 |
| momentum | 0.9 | 0.9 | 0.9 |
| warmup_epochs | 5 | 5 | 0 |
| warmup_learning_rate | $1e-6$ | $1e-6$ | N/A |
| optimizer | Adam | Adam | SGD |
| learning_rate_policy | Cosine decay | Cosine decay | Cosine decay |
| $\alpha$ (weight for $\mathcal{L}_{\text{text}}$) | 1 | 1 | 1 (for +Text-Attr) |

## G    USE OF LARGE LANGUAGE MODELS (LLMs)

Large Language Models (LLMs) were used in a limited manner, primarily to review the constructed hierarchy and to assist with minor tasks such as translation and typo correction.

