# OpenReview forum: "Free-Grained Hierarchical Recognition"
_ICLR.cc/2026/Conference — ICLR 2026 Conference Withdrawn Submission_

### Official Review · Reviewer_Pbd7 · 2025-10-17

**Soundness:** 1
**Presentation:** 2
**Contribution:** 2
**Rating:** 2
**Confidence:** 5

**Summary:**

This work introduced and studied the new task of free-grained hierarchical recognition, where supervision is done with varying label granularity but the model is required to predict the full taxonomy from training data with mixed labels.

To study and facilitate this newly proposed task, a new benchmark ImageNet-Free was curated from ImageNet to make ImageNet images have 3-level consistent label hierarchy.

Subsequently, free-grained learning framework based on pseudo attribute labels and contrastive losses are used to train a free-grained hierarchical recognition model.

Lastly, experiments are conducted on sufficient benchmarks.

**Strengths:**

- The paper is generally well-written
- The presentation is clear and intuitive. But it is a bit too dense.
- The curation of 3-level consistent hierarchy for ImageNet is a good contribution. Although the reviewer is  not convinced by the claims about its motivation and the usage in this work, having such consistent hierarchy is useful to train hierarchical recognition model.
- The experiments are sufficient and extensive.

**Weaknesses:**

**------ Primary weakness ------**

**(1) Is hard-organizing all concepts to 3-level hierarchy closer to real-world settings?:** The authors stressed a lot that the concept granularity varies significantly in real-world settings. Also, the authors stressed that the proposed free-grain learning task requires model to predict full taxonomy to reflect real-world concept variability and specificity.

However, in the newly proposed and curated ImageNet-Free dataset, the authors hard-organized concepts to 3-level hierarchy (basic, subordinate, fine-grained). Such structure is not realistic at all. In real-world setting: (a) Some concepts can have a deep taxonomy tree, such as Animals like Dog or Bird can all be organized under a 7-level hierarchy following the tree of life (species, genus, family, order, class, phylum, kingdom); (b) However, some concepts are like a Toy Teddy Bear can only be organized to a shallow hierarchy, like Toy Teddy Bear, Toy, Entity (root).

So, in real-world settings, the depth of semantic taxonomy should vary. This is natural. In addition, it can have multiple paths, like WordNet, based on the categorical criteria (category, material, usage, user case, etc), behind the taxonomy. Based on these, the reviewer does not find the proposed dataset and strategy realistic, practical and natural at all. To this end, the proposal of ImageNet-Free actually conflicts with the research goal claim of being closer to real-world settings. Actually, the proposed ImageNet-free seems artificial, and customized for the proposed task free-grain learning.

**------ Critical weakness ------**

**(2) Lack of proper citation to original technique innovation:** For the “proposed” technique “Text-Guided Pseudo Attributes (Text-Attr)” from Line#267 to Line#311, the authors neither include reference citation nor description crediting the literature where this technique was originally proposed.

**First, the attributes acquisition technique using large VLMs (or VQA +LLMs) are originally proposed by FineR [1]. Second, using such pseudo-labelled attributes as auxiliary supervision for training to improve large VLM’s fine-grained recognition performance is proposed  and used in Finedefics [2]. To this end, the techniques of “Text-Guided Pseudo Attributes (Text-Attr)” are not novel and originally from this work.**

**However, the authors did not cite or mention where the “proposed” Text-Guided Pseudo Attributes (Text-Attr) is originally from. The authors should appropriately acknowledge prior research contributions rather than implying that this innovation originates solely from their proposed method.**

**(3) Lack of discussion and comparison with highly-related work for technical details:** What is the difference between the taxonomy-aligned contrastive loss proposed in this work and the Attribute-Category Contrastive (ACC) loss proposed in Finedefics [2]? The ideas and implementation look similar. **Without discussion and comparison, the reviewer is not sure about the novelty of this component.**

[1] Liu, M., Roy, S., Li, W., Zhong, Z., Sebe, N., & Ricci, E. (2024). Democratizing fine-grained visual recognition with large language models. In ICLR, 2024.

[2] He, H., Li, G., Geng, Z., Xu, J., & Peng, Y. (2025). Analyzing and boosting the power of fine-grained visual recognition for multi-modal large language models. In ICLR 2025.

**Questions:**

**------ Questions ------**

- **(4)** Regarding the evaluation metrics, hierarchical precision, recall and F-score should also be considered to measure the mistake severity, see formulation established in Section 3.2.1  of [3].
- **(5)** Having a consistent categorical hierarchy of a dataset is indeed useful for downstream applications. In this work, the authors curated a 3-level consistent hierarchy for ImageNet. However, such idea is not new. Actually, both CHILS [4] and SHINE [5] leveraged LLMs to generate a 3-level hierarchy for ImageNet to improve zero-shot classification accuracy at different granularity level. Comparing to the LLM-generated hierarchy, what’s the advantage of the manually curated hierarchy? The author emphasized the importance of the definition of the hierarchy being under a cognitive psychology theoretical framework. Yet, when using LLM to automatically generate hierarchies might also be aligned with human cognitive common senses, because this is how we mention or describe the concepts in human daily life; this can be leveraged from the world knowledge encoded in modern LLMs.
- **(6)**  What’s the curation detail of ImageNet-Free? How long does it take? Where and how are the human annotator instructed and employed?
- **(7)** What would be a practical real-world use case of free-grained hierarchical recognition?

[3] Zhao, H., Puig, X., Zhou, B., Fidler, S., & Torralba, A. (2017). Open vocabulary scene parsing. In ICCV, 2017.

[4] Novack, Z., McAuley, J., Lipton, Z. C., & Garg, S. (2023, July). Chils: Zero-shot image classification with hierarchical label sets. In ICML, 2023.

[5] Liu, M., Hayes, T. L., Ricci, E., Csurka, G., & Volpi, R. (2024). Shine: Semantic hierarchy nexus for open-vocabulary object detection. In CVPR, 2024.

---

### Official Review · Reviewer_UJXJ · 2025-10-31

**Soundness:** 3
**Presentation:** 4
**Contribution:** 2
**Rating:** 4
**Confidence:** 4

**Summary:**

This paper introduces Free-Grain Learning, a new setting for hierarchical image classification where training labels can appear at different levels of granularity (e.g., “Bird”, “Bird of prey”, “Bald eagle”).
The authors further propose ImageNet-Free, a large-scale benchmark derived from ImageNet with a consistent three-level taxonomy (basic–subordinate–fine), and simulate realistic mixed-granularity supervision using CLIP confidence.

Two baseline methods are presented:
- Text-Attr, which leverages pseudo attributes from vision-language models for semantic guidance;
- Taxon-SSL, a taxonomy-guided semi-supervised learning method.

**Strengths:**

1. Realistic and well-motivated problem setting.
The free-grain supervision assumption captures real-world annotation variability and extends hierarchical classification to a more practical domain.

2.	Methodological clarity and practicality.
The two proposed approaches, Text-Attr and Taxon-SSL, are simple and conceptually clear. Their combination shows complementarity between semantic and visual cues.

3. Well written and clearly presented.
The manuscript is clearly structured, easy to follow, and provides sufficient implementation details. Figures and tables are well designed, contributing to overall readability.

**Weaknesses:**

Overall, a dataset or benchmark paper must demonstrate that it is not merely a reorganization of existing data but a contribution with the capability to advance scientific research. Specifically, it should quantitatively show its strengths in **representativeness** (accurately reflecting the target scenarios or real-world distributions of the research problem), **structural consistency** (ensuring that labels, hierarchical structures, and annotation schemes are logically coherent and cognitively meaningful), and **task distinctiveness and necessity** (proving that the newly defined task is not a simple renaming of existing ones but introduces genuinely new challenges or behavioral patterns). It should also verify the **correlation with real-world or downstream tasks**, showing that performance on the benchmark can predict, explain, or improve results in practical applications. Finally, it must include **challenging and sufficient baseline experiments** demonstrating that existing methods struggle on this benchmark, thereby establishing its scientific value and necessity. However, this paper falls short in several of the above aspects, details are as follows.
## Major:

### 1. Dataset analysis is insufficient.
Although the paper positions ImageNet-Free (datasets and benchmark) as its core contribution, it lacks a systematic quantitative analysis demonstrating its advantage over existing datasets for real world fine-grained recognition problems.
For example:
	•	No comparison between models trained on original ImageNet vs. ImageNet-Free on real-world test data.
	•	The assumption that CLIP-based pruning realistically mimics human annotation is not validated.
### 2.	Limited comparative evaluation.
The experiments only include HRN and H-CAST as baselines, but omit related paradigms such as Fine-grained classification with missing labels, or hierarchical semi-supervised methods, leaving the evaluation incomplete.
### 3.	Unverified realism of CLIP-based simulation.
The dataset construction relies heavily on CLIP confidence to mimic annotator uncertainty, yet no quantitative or human study supports this. The generated label sparsity patterns might not correspond to true annotation behavior.

## Minor

Related work discussion incomplete.
The task setting is conceptually close to partial-label learning on fine-grained data, and the use of “shared attributes” has been explored in fine-grained recognition literature such as [1][2], etc.
However, these connections are not explicitly discussed.

[1] Xu, W., Xian, Y., Wang, J., Schiele, B., & Akata, Z. (2022). Vgse: Visually-grounded semantic embeddings for zero-shot learning. CVPR 2022.

[2] Jiang, H., Sun, Z., & Tian, Y. (2024). Navigating real-world partial label learning: unveiling fine-grained images with attributes. AAAI 2024.

**Questions:**

Analysis see weaknesss.
1. **Quantitative validation of the dataset.**
   Have you conducted any quantitative analysis to show that *ImageNet-Free* better reflects real-world fine-grained recognition scenarios compared to the original ImageNet? For example, comparisons in granularity distribution, class similarity, or model performance transferability could strengthen the dataset’s representativeness.

2. **Verification of CLIP-based pruning.**
   How do you justify the assumption that CLIP confidence accurately simulates human annotation uncertainty? Was any manual inspection, human agreement study, or statistical validation performed to support this?

3. **Evaluation scope and baselines.**
   Why are paradigms such as partial-label learning or fine-grained classification with missing labels not included as baselines? Including these would provide a more comprehensive understanding of how existing approaches perform under the free-grain setting.

4. **Correlation to real-world or downstream tasks.**
   Does performance on *ImageNet-Free* correlate with performance on real-world datasets where annotation granularity varies naturally (e.g., iNaturalist, or weakly labeled wildlife datasets)? Such analysis would strengthen the benchmark’s practical value.

---

### Official Review · Reviewer_n9Be · 2025-10-31

**Soundness:** 3
**Presentation:** 3
**Contribution:** 3
**Rating:** 8
**Confidence:** 4

**Summary:**

The paper introduces free-grain learning, where image label granularity varies. To address the limitations of prior benchmarks, the authors introduce ImageNet-Free, a large-scale dataset with a consistent three-level hierarchy grounded in cognitive principles. They propose two main solutions: Text-Attr, which uses VLM-derived descriptions for semantic guidance, and Taxon-SSL, using visual consistency. The results show that the proposed method performs well.

**Strengths:**

The concept of free-grain learning is highly significant and addresses a realistic weakness of prior work. To this end, the paper also provides a strong benchmark.

The proposed method is outperforming existing methods.

The motivation is valid and novel. The paper is well-written.

**Weaknesses:**

Check the questions

**Questions:**

In line 269, what does the paper mean by visual attributes remain consistent? consistent across what? Pembroke images? or across different levels of hierarchy? because, for example, tail length can not stay consistent throughout different levels of hierarchy.


The symbol in equation 2 is not well defined and explained.

Line 313, what is CHMatch and what is its contrastive loss? (There is no citation provided, and no extra details about the method)

The paper includes a method to generate a caption. How does the quality of the generated captions affect the method?

The paper discusses pseudo attributes; however, there is no analysis and a clear definition provided.

---

### Official Review · Reviewer_YC4F · 2025-11-02

**Soundness:** 3
**Presentation:** 3
**Contribution:** 3
**Rating:** 6
**Confidence:** 4

**Summary:**

The paper introduces free‑grained learning: training hierarchical image classifiers when supervision varies in granularity across instances (e.g., only Bird for distant photos, but Bald eagle for close‑ups). To support this, the authors curate ImageNet‑Free, a large, three‑level taxonomy (basic → subordinate → fine‑grained) derived from ImageNet and grounded in cognitive psychology; they also provide analogous variants for iNat21‑mini and CUB, plus synthetic free‑grained versions of CUB and Aircraft. Methods include (i) Text‑Attr, which aligns image features to pseudo‑attributes extracted from VLM‑generated descriptions, and (ii) Taxon‑SSL, a taxonomy‑aware semi‑supervised approach; a simple consistency‑based stopping rule is used for fine‑grained inference. On ImageNet‑Free and iNat21‑mini‑Free, the proposed approaches substantially mitigate the large performance drop seen when strong hierarchical baselines are trained with mixed‑granularity labels.

**Strengths:**

1. Clear motivation and dataset curation. The paper diagnoses WordNet’s hierarchy issues (variable depths 5–19, multi‑path classes like minivan and teddy bear) and reconstructs a three‑level taxonomy aligned with basic/subordinate/fine‑grained categories, leading to ImageNet‑Free spans 20/127/505 classes with 645,480 train and 25,250 test images.

2. Simple, model‑agnostic methods with measurable gains. Text‑Attr (LLM descriptions ⇒ CLIP text embeddings ⇒ contrastive alignment) and Taxon‑SSL (taxonomy‑aligned contrastive loss) are straightforward to reproduce from the provided formulas (Eqs. 1–4, pp. 6–7). They significantly reduce the free‑grained drop compared to HRN/H‑CAST.

3. Careful evaluation protocol and metrics. The paper uses per‑level top‑1, Full‑Path Accuracy (FPA), and TICE (tree‑inconsistency rate), and quantifies how free‑grain supervision hurts H‑CAST (e.g., −39.8 pp FPA on iNat21‑mini‑Free)

4. Insightful analyses. Class‑wise analysis shows Text‑Attr excels under sparser fine labels, while Taxon‑SSL benefits when moderate labels exist (Fig. 8); consistency‑based stopping yields deeper yet consistent predictions (Fig. 10).

**Weaknesses:**

1. The semantic pruning rule is tied to CLIP/BioCLIP correctness/confidence (Sec. 3.2), then Text‑Attr uses CLIP’s text encoder for supervision (Sec. 4.2), creating a potential method–data coupling that could advantage Text‑Attr on ImageNet‑Free. This risk is not quantified.

2. The abstract claims mixed‑granularity labels “reflect human annotation behavior,” but no human validation is provided; realism is inferred from CLIP/BioCLIP behavior and qualitative examples.

3. The method section says it “extends CHMatch’s contrastive objective” to build $W^l$ and the intersection W (Eq. 3), but CHMatch itself isn’t introduced in background/related work, nor is it clear which parts (adaptive thresholding, teacher–student/EMA, strong/weak augs) are reused.

4. Eq. (4) sums over levels l=1..L, but the scored term only depends on W_{ij} and g(f(x)) with no explicit l. Readers can’t tell whether there are per-level projections g_l(f_l(x)) or whether the sum should be removed.

5. The text says “construct W^l from pseudo-label consistency,” but doesn’t specify: source of pseudo-labels (same model vs. EMA teacher), confidence thresholds and their schedule, weak/strong augmentation pairing, handling below-threshold items—precisely the knobs that determine behavior (and that CHMatch standardizes).

6. The paper rebuilds ImageNet into a three-level taxonomy and reports all results on this curated structure, but it did not evaluates on the unreconstructed WordNet hierarchy. Without this control, it’s hard to disentangle gains due to the proposed methods from gains due to taxonomy cleaning/simplification, and it weakens claims about robustness to real-world hierarchical noise (uneven depth, multi-parent classes). As a result, external validity is limited: we don’t know whether the methods still help when operating on the original, messy hierarchy practitioners actually face.

**Questions:**

Please refer to the weaknesses.

---

### Note · Authors · 2025-11-14

I have read and agree with the venue's withdrawal policy on behalf of myself and my co-authors.